# Observations of linear aggregation behavior in rotifers (*Brachionus calyciflorus*)

**Shuang-Huai Cheng** [1][☯]*, **Hai-Ying Zhang**[1][☯], **Ming-Yue Zhu**[2], **Li Min Zhou**[1], **Guo-Hui Yi**[1], **Xiao-Wen He**[1], **Jin-Yan Wu**[1], **Jin-Lei Sui**[1], **Hua Wu**[1], **Shi-Jiao Yan**[3,4]*, **Yun-Xia Zhang**[1,3]*, **Chuan-Zhu Lv**[3,4,5,6,7]*

**1** Public Rsearch Lab, Hainan Medical University, Haikou, Hainan, China, **2** Department of Basic Medicine and Life Sciences, Hainan Medical University, Haikou, Hainan, China, **3** Research Unit of Island Emergency Medicine, Chinese Academy of Medical Sciences (No. 2019RU013), Hainan Medical University, Haikou, Hainan, China, **4** School of Public Health, Hainan Medical University, Haikou, Hainan, China, **5** Key Laboratory of Emergency and Trauma of Ministry of Education, Hainan Medical University, Haikou, Hainan, China, **6** Department of Emergency, Hainan Clinical Research Center for Acute and Critical Diseases, The Second Affiliated Hospital of Hainan Medical University, Haikou, Hainan, China, **7** Emergency and Trauma College, Hainan Medical University, Haikou, Hainan, China

☯ These authors contributed equally to this work.
* shuanghuaicheng2018@hainmc.edu.cn (SC); yanshijiao@hainmc.edu.cn (SY); yunxiazhang@hainmc.edu.cn (YZ); lvchuanzhu677@126.com (CL)

**Data Availability Statement:** All relevant data are within the paper and its Supporting Information files.

**Funding:** This study was supported by the Hainan Medical College Talent Introduction Research start-

## Abstract

Linear aggregation is present in some animals, such as the coordinated movement of ants and the migration of caterpillars and spinylobsters, but none has been reported on rotifers. The rotifers were collected and clone cultured in the laboratory at 25 ± 1˚C, under natural light (light intensity ~130 lx, L:D = 14:10). The culture medium(pH = 7.3) was formulated as described by Suga et al., and rotifers were fed on the micro algae *Scenedesmus obliquus* grown in HB-4 medium to the exponential growth stage. When density was high (150 individuals ml$^{-1}$), the behavior of rotifers was observed using a stereo microscope (Motic ES-18TZLED). In this paper, linear aggregation in *Brachionus calyciflorus* was found for the first time, and experiments were carried out to verify the correlation between linear aggregation and culture density of *B. calyciflorus*. With the increase of density, the number of aggregations increase, the number of individuals in the aggregation increased, and the maintenance time of the aggregation was also increased. Therefore, we speculate that the formation of aggregates is related to density and may be a behavioral signal of density increase, which may transmit information between density increase and formation of dormant eggs.

## Introduction

Little research has been done on rotifer behavior. Mating behavior of the rotifer was described: *Epiphanes senta* [1], *Brachionus plicatilis* [2] and *Brachionus calyciflorus* [3]. Mating behavior of *B. plicatilis* [2] and *B. calyciflorus* [3] was described as five phases: encounter, circling, coronal localization, sperm transfer, and dissociation. The oviposition behavior of the littoral rotifer *Euchlanis dilatata* was investigated by Elizabeth in 1989 [4]. Researchers investigated the

up project (XRC190013); The National Natural Science Foundation of China (Grant No. 31860035). The funders provides financial support.

effect of toxic substances treatments on swimming speed of *Brachionus calyciflorus*: bromate [5] and $TiO_2$ nanoparticle [6]. Kim et al. investigated the effects of five external factors on movement in females of *Brachionus plicatilis* sensu stricto: food limitations, temperature, salinity, predator, and un-ionized ammonia [7]. The effect of UV-B radiation on the feeding behavior of the rotifer *Brachionus plicatilis* was displayed by Feng et al [8]. But there is no report on the behavior of linear aggregation in rotifer.

Social animals employ certain types of organized behaviors to achieve tasks that are difficult to accomplish as individuals. For example, ants are known to form a chain [9] to carry heavy objects. According to the Daily Mail [10], some 136 caterpillars formed an orderly queue and wriggled head-to-tail across a road in single file. Similar group behaviors are also exhibited spiny lobsters, which form a chain in mass migrations, this behavior may serve a defensive function [11]. However, we are not aware of any other reports that rotifers, an asocial animal, form daisy chain-like associations. We report that rotifers form daisy chain-like associations, providing evidence that this organized social behavior is present among invertebrates living in the zooplankton.

## Materials and methods

Freshwater rotifers, *Brachionus calyciflorus* were collected from a small pond on the campus of Hainan Medical University, Longhua District, Haikou City, Hainan Province in China (Longitude:110.19, Latitude:19.58). The rotifers were collected and clone cultured in the laboratory at $25 \pm 1°C$, under natural light (light intensity ~130 lx, L:D = 14:10). The culture medium (pH = 7.3) was formulated as described by Suga et al [12]. Rotifers were fed on the microalgae *Scenedesmus obliquus* grown in HB-4 medium [13] to the exponential growth stage. When density was high (150 individuals $ml^{-1}$), the behavior of rotifers were observed using a stereo microscope (Motic ES-18TZLED).

Under the microscope, the rotifers, *B. calyciflorus* formed daisy chain-like aggregations of two to six individuals, most of which consisted of two or three individuals (Fig 1). The linear aggregation pattern was showed in Fig 2. The daisy chain forms frequently moved quickly in one direction. Notably, only the last rotifer in the line had eggs (dormant or mictic).

We conducted two experiments to examine the formation of linear aggregations of rotifers. In the first experiment, *B. calyciflorus* were observed at different densities (from 10 to 400

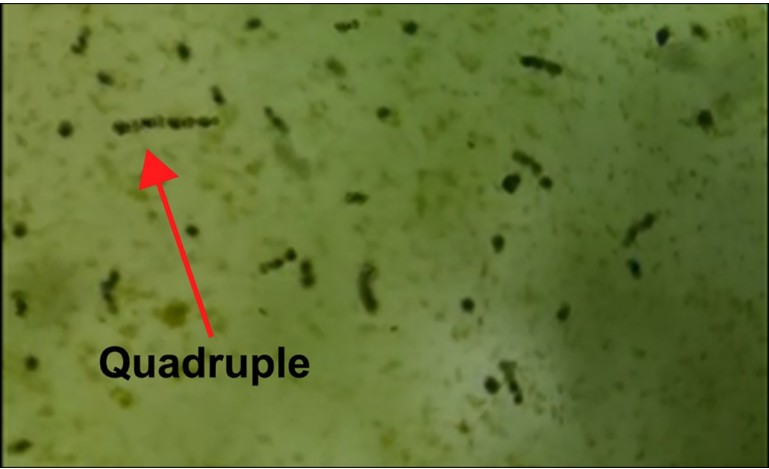

**Fig 1. The phenomenon of linear aggregation observed by microscope.**

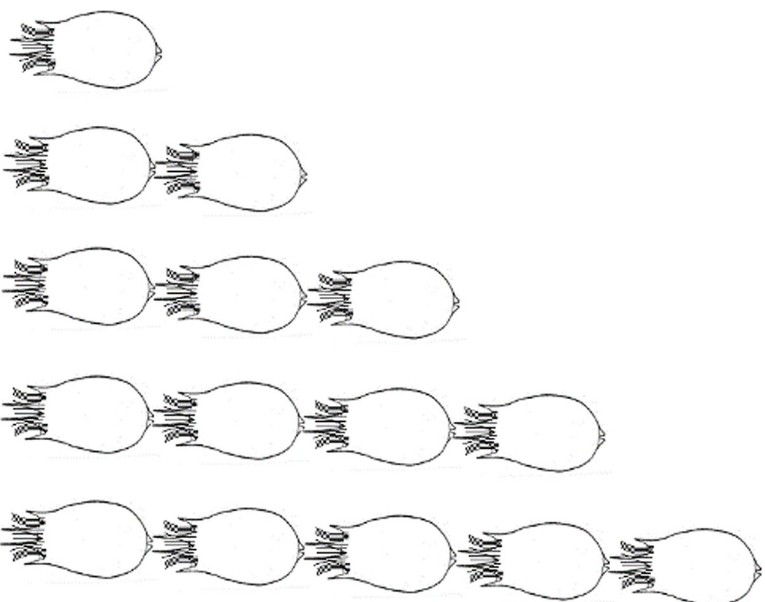

**Fig 2. The linear aggregation pattern of *B. calyciflorus*.**

individuals 4ml$^{-1}$) (Table 1). The individuals used in the experiments were selected at random and placed into 4 ml of culture medium to observe the number of aggregations and the number of individuals in each aggregation. The results are recorded in Table 1. In the second experiment, *B. calyciflorus* multiples of two (to a total of 32 individuals) were placed into 1 ml of culture medium and the number of aggregations formed, and the time over which they persisted, was recorded for ten minutes each density experiment (Table 2). At last, we add the high density *B. angularis* into the low density *B. calyciflorus*, the *B. calyciflorus* also form daisy chain-like aggregations of two individuals.

## Results and discussion

The results indicate that the formation of linear aggregations increases with an increase in the density of rotifers. The first linear aggregation formed at a density of 50 individuals 4 ml$^{-1}$ in the first experiment, and at 16 individuals ml$^{-1}$ in the second experiment. The first linear aggregation of three individuals occurred at a density of 32 individuals ml$^{-1}$ in the second experiment. The longest duration of a linear aggregation of two individuals was 481 seconds in the second experiment at a density of 32 individuals ml$^{-1}$. In general, linear aggregations were more stable at higher densities.

Linear aggregations of fossil arthropods were found by Hou et al. They believe that the formation of the stable linear structure found for this species is unique among arthropods,

**Table 1. Experimental results of correlation between density and linear aggregation.**

| Number of *B. calyciflorus* (Individuals/4 ml) | | 10 | 50 | 100 | 200 | 300 | 400 |
|---|---|---|---|---|---|---|---|
| Number of multiple connectomes | dimer | 0 | 2 | 3 | 9 | 8 | 9 |
| | triplet | 0 | 0 | 1 | 1 | 2 | 3 |
| | quadruple | 0 | 0 | 0 | 0 | 1 | 2 |
| | quintuple | 0 | 0 | 0 | 0 | 0 | 1 |

**Table 2. Number and duration of dimer occurrences in ten minutes.**

| Test order | Individuals/1 ml | Number of dimer occurrences | Duration of the dimer(seconds) | | | | | | | |
|---|---|---|---|---|---|---|---|---|---|---|
| 1 | 2 | 0 | | | | | | | | |
| 2 | 4 | 0 | | | | | | | | |
| 3 | 6 | 0 | | | | | | | | |
| 4 | 8 | 0 | | | | | | | | |
| 5 | 10 | 0 | | | | | | | | |
| 6 | 12 | 0 | | | | | | | | |
| 7 | 14 | 0 | | | | | | | | |
| 8 | 16 | 0 | | | | | | | | |
| 9 | 18 | 1 | 23 | | | | | | | |
| 10 | 20 | 1 | 25 | | | | | | | |
| 11 | 22 | 1 | 142 | | | | | | | |
| 12 | 24 | 5 | 16 | 19 | 5 | 3 | 3 | | | |
| 13 | 26 | 2 | 7 | 5 | | | | | | |
| 14 | 28 | 3 | 288 | 70 | 46 | | | | | |
| 15 | 30 | 6 | 7 | 49 | 6 | 6 | 5 | 17 | | |
| 16* | 32 | 8 | 42 | 151 | 21 | 183 | 29 | 24 | 481 | 95 |

* Triplets occurrence for 7 seconds.

whether fossil or living [14]. Vannier et al. also found linear aggregations of fossil arthropods in Morocco in 2019 [15]. However linear aggregations do not seem to be common among living invertebrates. We are the first to record linear aggregations of living rotifers in culture.

Studies have shown that bacteria perceive and respond to changes in population density through a communication system between cells, which is called quorum sensing. In this system, small molecular hormones called 'autoinducers' are produced, released, detected and responded to by bacteria. With bacterial population growth, the concentration of autoinducers released by bacteria also increases. When the concentration of these signal molecules reaches a critical value, bacteria can detect their presence, then amplify the signal, resulting in the expression of target genes [16]. Whiteley et al reviewed progress in and promise of bacterial quorum sensing research. Bacterial communicate with each other and to coordinate their activities by quorum sensing signals. The researches of genetics, genomics, biochemistry, and signal diversity of QS were also summarized [17]. The phenomenon of linear aggregations in *B. calyciflorus* occurs at high population densities. This may be a behavior signal, which stimulates the expression of specific genes in *B. calyciflorus*. This could result in the appearance of males and mictic females, leading to mating and the production of dormant eggs. Further research is needed to understand the molecular mechanism that stimulates the production of dormant eggs when *B. calyciflorus* forms linear aggregations.

Further research is also required to understand whether linear aggregations will occur in other rotifer species, or for other cryptic species of *B. calyciflorus*, as density increases during the culturing process. At present, this phenomenon has not been observed in cultures of *B. angularis*, *B. caudatus*, *B. diversicornis*, *B. forficula*, *B. quadridentatus*, *B. urceolaris*, *B. rubens*, *B. falcatus*, or *B. patulus* in our laboratory. In high-density cultures of cryptic species of *B. calyciflorus*, the formation of linear aggregations could also provide a warning that density is too high, providing visual guidance for the large-scale culture of this species.

Linear aggregation may give predators the illusion of size and avoid prey, just like spiny lobsters [11]. Whether the linear aggregation of *B. calyciflorus* is related to predation pressure

needs to be confirmed by relevant experiments. Whether the linear aggregation of *B. calyciflorus* causes the aggregated rotifers to save motor energy and use more energy for reproductive activities remains to be verified by further experiments.

*B. calyciflorus* and *B. havanaensis* exhibited costly and effective defences after induction by predator (*Asplanchna brightwelli*) infochemicals [18]. Is linear aggregation of *B. calyciflorus* also caused by some infochemicals? And does linear aggregation of *B. calyciflorus* product new infochemicals? The answers need more deeply researches.

## Supporting information

**S1 Movie. Video of linear aggregates of *B. calyciflorus* at high density.**
(MP4)

## Author Contributions

**Conceptualization:** Ming-Yue Zhu, Shi-Jiao Yan, Yun-Xia Zhang.

**Data curation:** Shuang-Huai Cheng, Li Min Zhou, Guo-Hui Yi, Xiao-Wen He, Jin-Lei Sui, Hua Wu.

**Formal analysis:** Hai-Ying Zhang.

**Funding acquisition:** Jin-Yan Wu.

**Investigation:** Hai-Ying Zhang, Jin-Yan Wu.

**Methodology:** Shi-Jiao Yan, Yun-Xia Zhang.

**Project administration:** Yun-Xia Zhang.

**Resources:** Li Min Zhou, Guo-Hui Yi, Xiao-Wen He, Jin-Lei Sui, Hua Wu.

**Supervision:** Chuan-Zhu Lv.

**Writing – original draft:** Shuang-Huai Cheng.

**Writing – review & editing:** Ming-Yue Zhu, Shi-Jiao Yan, Chuan-Zhu Lv.

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
