## [Decision Letter · Decision Letter 0]

30 Mar 2021

PONE-D-20-32010

Observations of linear aggregation Behavior in Rotifers (Brachionus calyciflorus)

PLOS ONE

Dear Dr. Cheng,

Thank you for submitting your manuscript to PLOS ONE. After careful consideration, we feel that this manuscritpt has merit but does not fully meet PLOS ONE’s publication criteria as it currently stands. Therefore, we invite you to submit a revised version of the manuscript that addresses the points raised during the review process.

We look forward to receiving your revised manuscript.

Kind regards,

Ram Kumar, Ph.D.

Academic Editor

PLOS ONE

Additional Editor Comments:

Dear Author

I have been able to get your manuscript reviewed by two reviewers. Both the reviewers suggest minor revision before your manuscript can be accepted for publication. I am also of the opinion that the manuscript need revision, and consequences of linear aggregation behaviour in view of escape from copepod and fish predation and fitness of egg carrying zooplankton like Brachionid rotifers should be elaborated. Formatting should be done carefully and recent published information should also be discussed and elaborated.

Journal Requirements:

"This study was supported by the Hainan Medical College Talent Introduction

Research start-up project (XRC190013); The National Natural Science Foundation of

China (Grant No. 31860035). The funders provides financial support."

"The funders had no role in study design, data collection and analysis, decision to publish, or preparation of the manuscript"

Reviewers' comments:

Reviewer's Responses to Questions

**Comments to the Author**

1. Is the manuscript technically sound, and do the data support the conclusions?

Reviewer #1: Yes

Reviewer #2: Yes

2. Has the statistical analysis been performed appropriately and rigorously? 

Reviewer #1: N/A

Reviewer #2: N/A

3. Have the authors made all data underlying the findings in their manuscript fully available?

Reviewer #1: Yes

Reviewer #2: Yes

4. Is the manuscript presented in an intelligible fashion and written in standard English?

Reviewer #1: Yes

Reviewer #2: Yes

5. Review Comments to the Author

Reviewer #1: The manuscript entitled "Observations of linear aggregation Behavior in Rotifers (Brachionus calyciflorus)" discusses linear aggregation in Brachionus calyciflorus upon increase of density and such behavioral pattern among asocial animal like rotifers is an interesting finding. The manuscript may be accepted with minor corrections. The authors need to correct various formatting issues, spelling and grammatical mistakes e.g. " mL-1" etc. Similarly, references also need formatting and

more latest references should be included such as Nature, 2017, 551,313 on bacterial quorum sensing etc.

Reviewer #2: The manuscript entitled "Observations of linear aggregation Behavior in Rotifers (Brachionus calyciflorus)" discusses previously undocumented, linear aggregation in rotifer Brachionus calyciflorus in response to increased culture density. The novel findings are well documented and presented. The manuscript may be accepted with few minor corrections. The author shall review and correct various formatting, grammatical and spelling errors and arrange the references correctly.

6. PLOS authors have the option to publish the peer review history of their article (what does this mean?). If published, this will include your full peer review and any attached files.

Reviewer #1: No

Reviewer #2: No

---

## [Author Response · Author response to Decision Letter 0]

6 May 2021

Dear Editor, 

Thank you for carefully reviewing our manuscript previously titled “Observations of linear aggregation Behavior in Rotifers (Brachionus calyciflorus)” for possible publication in the PLOS ONE. We are grateful to you and your reviewers for their constructive critique. We have revised the manuscript, highlighting our revisions in red and have attached point-by-point responses detailing how we have revised the manuscript in response to the reviewers' comments below.

We added reviewe of rotifer behaviors, and cited some latest references, and corrected various formatting issues, spelling and grammatical mistakes e.g. and arranged the references correctly.

Thank you for your consideration and further review of our manuscript. Please do not hesitate to contact us with any further questions or recommendations.

Yours Sincerely,

shuanghuaicheng2018@hainmc.edu.cn (SC); yanshijiao@hainmc.edu.cn (SY); yunxiazhang@hainmc.edu.cn (YZ); lvchuanzhu677@126.com (CL).

 

Reviewer Comments:

Reviewer #1: The manuscript entitled "Observations of linear aggregation Behavior in Rotifers (Brachionus calyciflorus)" discusses linear aggregation in Brachionus calyciflorus upon increase of density and such behavioral pattern among asocial animal like rotifers is an interesting finding. The manuscript may be accepted with minor corrections. The authors need to correct various formatting issues, spelling and grammatical mistakes e.g. " mL-1" etc. Similarly, references also need formatting and more latest references should be included such as Nature, 2017, 551,313 on bacterial quorum sensing etc.

Response:

Thank you for your reviewing of our article and giving suggestions. We have modified it according to your suggestions and submitted the revised manuscript. Including the spelling of words, grammar mistakes, the format of the references and so on. And add some latest references including such as Nature, 2017, 551,313 on bacterial quorum sensing etc.

Reviewer #2: The manuscript entitled "Observations of linear aggregation Behavior in Rotifers (Brachionus calyciflorus)" discusses previously undocumented, linear aggregation in rotifer Brachionus calyciflorus in response to increased culture density. The novel findings are well documented and presented. The manuscript may be accepted with few minor corrections. The author shall review and correct various formatting, grammatical and spelling errors and arrange the references correctly.

Response:

Thank you for your reviewing of our article and giving suggestions. We have modified it according to your suggestions and submitted the revised manuscript. Including the spelling of words, grammar mistakes, the format of the references and so on.

---

## [Decision Letter · Decision Letter 1]

6 Aug 2021

Observations of linear aggregation Behavior in Rotifers (Brachionus calyciflorus)

PONE-D-20-32010R1

Dear Dr. Cheng,

We’re pleased to inform you that your manuscript has been judged scientifically suitable for publication and will be formally accepted for publication once it meets all outstanding technical requirements.

Kind regards,

Arumugam Sundaramanickam, PhD

Academic Editor

PLOS ONE

Additional Editor Comments (optional):

Reviewers' comments:

Reviewer's Responses to Questions

**Comments to the Author**

1. If the authors have adequately addressed your comments raised in a previous round of review and you feel that this manuscript is now acceptable for publication, you may indicate that here to bypass the “Comments to the Author” section, enter your conflict of interest statement in the “Confidential to Editor” section, and submit your "Accept" recommendation.

Reviewer #1: All comments have been addressed

2. Is the manuscript technically sound, and do the data support the conclusions?

Reviewer #1: Yes

3. Has the statistical analysis been performed appropriately and rigorously? 

Reviewer #1: N/A

4. Have the authors made all data underlying the findings in their manuscript fully available?

Reviewer #1: Yes

5. Is the manuscript presented in an intelligible fashion and written in standard English?

Reviewer #1: Yes

6. Review Comments to the Author

Reviewer #1: Authors have satisfactorily addressed all the points raised. Manuscript is now acceptable for publication in the Journal

7. PLOS authors have the option to publish the peer review history of their article (what does this mean?). If published, this will include your full peer review and any attached files.

Reviewer #1: No

---

## [Editor Report · Acceptance letter]

10 Aug 2021

PONE-D-20-32010R1 

Observations of linear aggregation Behavior in Rotifers (*Brachionus calyciflorus*) 

Dear Dr. Cheng:

I'm pleased to inform you that your manuscript has been deemed suitable for publication in PLOS ONE. Congratulations! Your manuscript is now with our production department. 

Kind regards, 

on behalf of

Professor Arumugam Sundaramanickam 

Academic Editor

PLOS ONE